

# Genome-wide identification and expression analysis of the ERF transcription factor family in pineapple (*Ananas comosus* (L.) Merr.)

Youmei Huang[1,*], Yanhui Liu[1,*], Man Zhang[1], Mengnan Chai[1], Qing He[1], Bello Hassan Jakada[1], Fangqian Chen[1], Huihuang Chen[1], Xingyue Jin[1], Hanyang Cai[1] and Yuan Qin[1,2]

[1] State Key Lab of Ecological Pest Control for Fujian and Taiwan Crops; Key Lab of Genetics, Breeding and Multiple Utilization of Crops, Ministry of Education; Fujian Provincial Key Lab of Haixia Applied Plant Systems Biology, College of Life Sciences, Fujian Agriculture and Forestry University, Fuzhou, Fujian Province, China

[2] State Key Laboratory for Conservation and Utilization of Subtropical Agro-Bioresources, Guangxi Key Lab of Sugarcane Biology, College of Agriculture, Guangxi University, Nanning, Guangxi Province, China

* These authors contributed equally to this work.

Corresponding authors
Hanyang Cai, cai-hanyang123@163.com
Yuan Qin, yuanqin@fafu.edu.cn

## ABSTRACT

Pineapple (*Ananas comosus* (L.) Merr.) is an important tropical fruit with high economic value. The quality and yield of pineapple will be affected by various environmental conditions. Under adverse conditions, plants can produce a complex reaction mechanism to enhance their resistance. It has been reported that the member of ethylene responsive transcription factors (ERFs) plays a crucial role in plant developmental process and stress response. However, the function of these proteins in pineapple remains limited. In this study, a total of 74 *ERF* genes *(AcoERFs)* were identified in pineapple genome, named from *AcoERF1* to *AcoERF74*, and divided into 13 groups based on phylogenetic analysis. We also analyzed gene structure, conserved motif and chromosomal location of *AcoERFs*, and the *AcoERFs* within the same group possess similar gene structures and motif compositions. Three genes (*AcoERF71*, *AcoERF73* and *AcoERF74*) were present on unanchored scaffolds, so they could not be conclusively mapped on chromosome. Synteny and *cis*-elements analysis of *ERF* genes provided deep insight into the evolution and function of pineapple *ERF* genes. Furthermore, we analyzed the expression profiling of *AcoERF* in different tissues and developmental stages, and 22 *AcoERF* genes were expressed in all examined tissues, in which five genes (*AcoERF13*, *AcoERF16*, *AcoERF31*, *AcoERF42*, and *AcoERF65*) had high expression levels. Additionally, nine *AcoERF* genes were selected for functional verification by qRT-PCR. These results provide useful information for further investigating the evolution and functions of ERF family in pineapple.

## INTRODUCTION

Plant growth and development, yield and quality are frequently affected by unfavorable environmental factors such as drought, salinity, high temperature and cold. In order to survive these stress conditions, plants have developed complex reaction mechanisms at the molecular, cellular, and system levels (*Mittler, 2006*; *Loudet & Hasegawa, 2017*). Gene co-expression at the transcriptional level is one of the most important ways to regulate biological processes. Transcription factors (TFs) play important roles in regulating the expression of functional proteins when plants are exposed to unfavorable environmental conditions (*Cui et al., 2016*). Among these TFs, ethylene responsive transcription factors (ERFs) play important roles in various biological processes, such as defense responses and hormonal signal transduction (*Agarwal et al., 2010*; *Sharma et al., 2010*; *Rashid et al., 2012*). Therefore, it is important to study the evolution and function of these genes in order to improve yield and the ability to resist the adverse environmental conditions of plant.

The AP2/ERF super-family is one of the largest family in plant, and the member of this super-family characterized by one or two conserved AP2/ERF domains of approximately 60 to 70 amino acids (*Sakuma et al., 2002*). It has been reported that the AP2/ERF domain contains tow conserved elements, YRG and RAYD element, and the core region of RAYD element has been predicted to form an amphipathic $\alpha$-helix (*Okamuro et al., 1997*). The three-dimensional (3D) structure of the AP2/ERF domain from AtERF1 shows that the domain is composed of a 3 anti-parallel $\beta$-sheets and an $\alpha$-helix (*Allen et al., 1998*). Furthermore, tryptophan (Trp) and arginine (Arg) residues of the $\beta$-sheet are in contact with DNA during transcription (*Sharma et al., 2010*). Based on the number of AP2/ERF domains and other DNA binding domains, the AP2/ERF super-family can be divided into three families:AP2, ERF and RAV (*Sakuma et al., 2002*; *Nakano et al., 2006*). Among the three families, only the AP2 family contains two conserved AP2/ERF domains, whereas ERF and RAV family contain just a single AP2/ERF domain (*Riechmann & Meyerowitz, 1998*). The members of the RAV family include an AP2/ERF domain and a B3domain (*Romanel et al., 2009*). Based on the ERF domain binding to DNA sequences, ERF family has been further classified into two major subfamilies: ERF (ethylene responsive transcription factors) subfamily and CBF/DREB (C-repeat binding factor/dehydration-responsive element binding factor) subfamily (*Nakano et al., 2006*). Proteins encoded by ERF subfamily genes bind to GCC box (AGCCGCC) (*Ohmetakagi & Shinshi, 1995*; *Hao, Ohmetakagi & Sarai, 1998*), whereas the CBF/DREB subfamily genes typically binds to the DRE and/or C-repeat (A/GCCGAC) (*Stockinger, Gilmour & Thomashow, 1997*; *Riechmann & Meyerowitz, 1998*). The residues at position 14 and 19 of ERF subfamily genes AP2/ERF domain is alanine (Ala) and aspartic acid (Asp), respectively, whereas DREB subfamily genes containing valine (Val) at position 14 and glutamine acid (Glu) at positions 19 (*Sakuma et al., 2002*).

ERFs are located downstream of the ethylene signaling pathway and regulate the transduction of ethylene as well as the trans-activation of certain transcription factors related to hormonal signal (*Fujimoto et al., 2000*; *Sharma et al., 2010*; *Yu et al., 2012*). In addition, ERF proteins are involved in various plant biological processes, such

as environmental stresses response (*Liu, White & Macrae, 1999*; *Rashotte et al., 2006*), beneficial symbiotic interaction (*Vernie et al., 2008*), as well as other developmental processes, such as leaf, flower and embryo development in some plants (*Elliott et al., 1996*; *Moose & Sisco, 1996*; *Boutilier et al., 2002*; *Vernie et al., 2008*). To date, ERF family has been reported in various plant species, such as *Arabidopsis thaliana* (*Sakuma et al., 2002*; *Nakano et al., 2006*), soybean (*Glycine max*) (*Li et al., 2005*; *Zhang et al., 2009*), rice (*Oryza sativa*) (*Cao et al., 2006*; *Nakano et al., 2006*; *Sharoni et al., 2011*; *Rashid et al., 2012*), cotton (*Gossypium barbadense* L.) (*Jin & Liu, 2008*; *Meng et al., 2010*), *Populus trichocarpa* (*Zhuang et al., 2008*), tomato (*Solanum tuberosum* L.) (*Sharma et al., 2010*; *Pan et al., 2012*), grape ((*Vitis vinifera* L.) (*Licausi et al., 2010*), cucumber (*Cucumis sativus* L.) (*Hu & Liu, 2011*) and tartary buckwheat (*Fagopyum Tataricum*) (*Liu et al., 2019*).

Pineapple (*Ananas comosus* (L.) Merr.) is one of the most important tropical fruit in the world (*Su et al., 2017*; *Xie et al., 2018*; *He et al., 2019*). The growth and development of pineapple is affected by various environmental conditions, such as drought, salt and cold stress (*He et al., 2019*). Additionally, pineapple is also an important monocotyledonous, and it can be considered as a proper model to study the monocot evolution (*Ming et al., 2015*; *He et al., 2019*). Recently, a comprehensive study of the pineapple genome provide a solid foundation for the study of pineapple gene functions (*Ming et al., 2015*; *Fang et al., 2016*; *Su et al., 2017*). Due to the great economic and research value of pineapple, it would be meaningful to make a further study ERF family in pineapple.

Previous study only proposed the existence of DREB subfamily in the ERF family of pineapple (*Chai et al., 2020*). However, detail information about the whole ERF family members in pineapple remains unexplored. In order to investigate the function and evolution of other ERF family members in pineapple, we conducted a detailed analysis of the entire ERF family. In this study, 74 pineapple *ERF* genes were identified and classified into 13 groups. Furthermore, we also conducted a systematic analysis including gene structure, motif compositions, chromosome distribution, phylogenetic and synteny analysis of each *AcoERF* gene. In order to further investigate the functions of *AcoERF* genes, we analyzed the expression profile of *AcoERF* genes in different tissues and stages. The data generated in this study will help to select the appropriate candidate genes for further functional studies during pineapple growth and development. This is of great significance to the investigation of pineapple stress response and variety improvement.

## MATERIALS & METHODS

### Genome-wide analysis of the ERF family in pineapple

The AP2/ERF amino acid sequences of pineapple and *Arabidopsis* were downloaded from Phytozome12 (https://phytozome.jgi.doe.gov/pz/portal.html) (*Goodstein et al., 2012*). The Hidden Markov Model (HMM) profiles of the AP2 domain (PF00847) and the B3 domain (PF02362) were obtained from PFAM database (http://pfam.sanger.ac.uk/) (*Finn et al., 2008*) as the queries for search predicted AP2/ERF proteins in the pineapple dataset using HMMER software 3.0 (http://hmmer.wustl.edu/) with a threshold of *e*-value$<e^{-5}$ (*Eddy, 2011*). BLAST searches were used to determine the predicted AP2/ERFs

in pineapple database with all the *Arabidopsis* AP2/ERFs as queries. All of the candidate genes were further examined by Simple Modular Architecture Research Tool (SMART, http://smart.embl-heidelberg.de/) (*Letunic & Bork, 2018*) to confirm the protein sequences derived from the selected pineapple AP2/ERF domains. ExPasy proteomics server (http://expasy.org/) (*Gasteiger et al., 2003*) was used to predicate the basic information of *AcoERF* proteins, including molecular weight (MW) and isoelectric points (*pI*).

### Gene ontology (GO) annotation

To investigate the putative function of the candidate *AcoERF* genes, the gene ontology (GO) annotations of the pineapple ERF family members were downloaded from Phytozome12 (https://phytozome.jgi.doe.gov/pz/portal.html) (*Goodstein et al., 2012*).

### Sequence alignment and phylogenetic analysis

Multiple sequence alignment of the AcoERF proteins was performed using DNAMAN (version 9) with the default parameters (*Wang, 2015*), and the diagram was visualized using ESPript 3.0 (http://espript.ibcp.fr/ESPript/cgi-bin/ESPript.cgi) (*Gouet, Robert & Courcelle, 2003*). To explore the evolution relationships of *ERF* gene family members in pineapple, the ERF amino acid sequences from pineapple and *Arabidopsis* were used. The multiple sequence alignments of all ERF proteins were performed by using MUSCLE (http://www.ebi.ac.uk/Tools/msa/muscle/) (*Edgar, 2004*), with the default parameters. Subsequently, the phylogenetic trees were constructed using MEGA6.0 software (http://www.megasoftware.net) (*Tamura et al., 2013*) via the Neighbor-joining (NJ) method with the following parameters: node robustness was detected using the bootstrap method, and the bootstrap was set to 1000 replications. Finally, the phylogenetic trees were generated by iTOL (https://itol.embl.de) (*Letunic & Bork, 2016*).

### Gene structure and conserved motif predictions

The exon/intron structures of *AcoERF* genes were obtained from the online program Gene Structure Display Server (GSDS, http://gsds.gao-lab.org/) (*Hu et al., 2015*). The conserved motifs in the AcoERF proteins sequences were investigated by Multiple EM for Motif Elicitation (MEME, http://meme-suite.org/tools/meme) (*Bailey et al., 2015*). The optimized parameters were as follows: maximum number of motifs was set to 20, the optimum width of motifs was set to 10-50 residues, and other options were set to default.

### Chromosomal localization and synteny analysis

The information of chromosome localization of all pineapple *ERF* genes was obtained from Phytozome12 (https://phytozome.jgi.doe.gov/pz/portal.html). And then, each genes was mapped on chromosomes using MapChart software based on positional information in the pineapple genome project (*Voorrips, 2002*). For synteny analysis, the sequence of rice and *Arabidopsis* ERF proteins were downloaded from Phytozome12. The potential anchors within pineapple, between pineapple and *Arabidopsis* or rice were searched by BLASTP with the score value of E$<$1e$^{-5}$ and top 5 matches (*Zhang et al., 2018*). The syntenic block of ERFs within pineapple was visualized using the Circos Program (version 0.69) (http://circos.ca/) (*Krzywinski et al., 2009*).

The syntenic blocks between pineapple and 2 representative plants were constructed using J. Craig Venter Institute (JCVI, https://github.com/tanghaibao/jcvi) (*Tanenbaum et al., 2010*). The Synonymous (*Ks*) and non-synonymous (*Ka*) values of AcoERF sequences were calculated using bio-pipeline-master (https://github.com/tanghaibao/bio-pipeline) (*Tang et al., 2008*).

## Expression analysis of *AcoERF* genes in different tissues

To investigate the expression patterns of the *AcoERF* genes in different development and growth processes of pineapple, the RNA-Seq data of pineapple roots, leaves, flowers, fruits (stage 1-6) were downloaded from the online website Pineapple Genomics Database (PGD, http://pineapple.angiosperms.org/pineapple/html/download.html) (*Ming et al., 2015*). The criterion of different stage of ovule, sepal, stamen and petal was referenced to the previously described methods (*Su et al., 2017*). Total RNA of these tissues was extracted using RNA extracted kit (Omega Bio-Tek, Shanghai, China) based on the supplier's instruction, and the samples were prepared based on the published method (*Chen et al., 2017*). The cDNA libraries for sequencing were constructed using the NEBNext Ultra$^{TM}$ RNA Library Prep Kit for Illumina (NEB). The reads were aligned to pineapple genome by the TopHat 2.0.0 software with default parameters (*Trapnell et al., 2012*). The Fragments Per Kilobase of transcript per Million fragments mapped (FPKM) value of each gene was calculated using Cuffdiff and Cufflinks software with default parameters. Finally, a heatmap of *AcoERF* expression profile was produced by the heatmaply R package (*Galili et al., 2018*).

## Analysis of *cis*-elements from *AcoERF* promoters

The promoter sequences of the *AcoERF* genes were downloaded from the Pineapple Genomics Database (http://pineapple.angiosperms.org/pineapple/html/index.html). And then, the stress- related cis-elements of AcoERF genes were predicted using the Plant Cis-Acting Regulatory Element (PlantCARE, http://bioinformatics.psb.ugent.be/webtools/plantcare/html/) (*Lescot et al., 2002*).

## Plant material and abiotic treatments

The pineapple materials were provided by the Qin Lab (Center for Genomics and Biotechnology, Fujian Agriculture and Forestry University, Fujian, China). The seedings were grown in a greenhouse at 25 °C, 16-h light/8-h dark photoperiod and a relative humidity of 70% for one month. And then the one-month-old pineapple plants were exposed to the following treatments: cold stress (4 °C), drought stress (350 mM Mannitol) and salt stress (150 mM NaCl). The samples of pineapple roots and leaves were collected after 6, 12, 24 and 48 h treatments. The one-month-old plants that without any stress treatment were used as controls in the expriment. The leaves and roots were harvested at the indicated time, and stored at −80 °C for subsequent analysis.

## RNA isolation and qRT-PCR analysis

Total RNA of the pineapple sample was extracted from roots and leaves using plant RNA extraction kit (OMEGA, Shanghai, China) according to the manufacturer's protocol. And then, total RNA was reverse-transcribed using AMV reverse transcriptase (Takara, Japan)

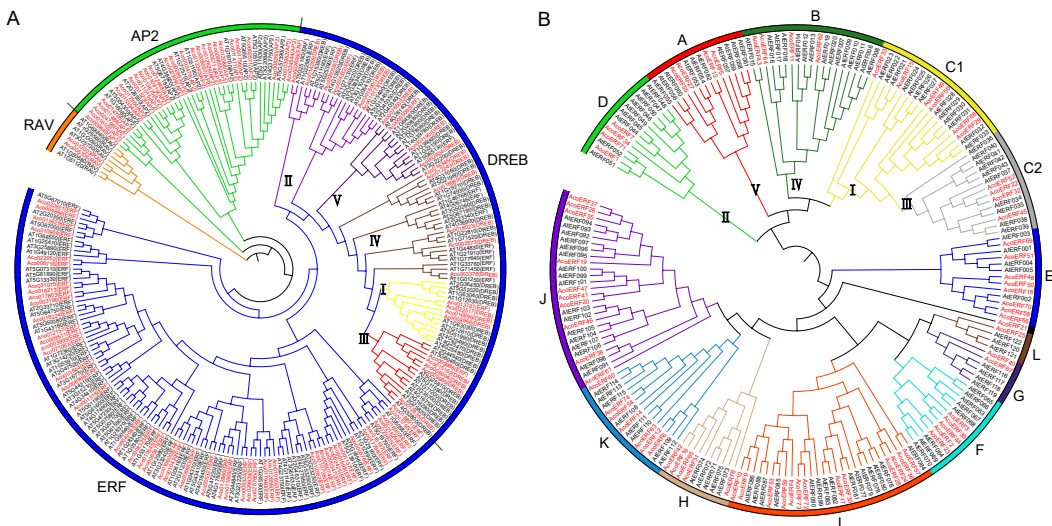

**Figure 1** **The evolutionary relationship of the *ERFs*.** (A) Unrooted phylogenetic tree representing relationship among *AP2/ERF* genes between pineapple and *Arabidopsis*; (B) Phylogenetic analysis of ERF proteins from pineapple and *Arabidopsis*. The genes in pineapple are marked in red, while those in *Arabidopsis* are marked in black. The pineapple DREB proteins are classified into five groups: I, II, III, IV and V.

based on the supplier's instruction. qRT-PCR analysis was conducted using SYBR Premix Ex Taq II system (Takara, Japan) and Bio-Rad Real-time PCR system (Foster city, CA, USA). The reactions were performed in a 20 μl volume containing 10 μl of 2× SYBR Premix, 8.2 μl of RNase free water, 1 μl of template, 0.4 μl of each specific primer. The primers used for RT-PCR were showed in Table S1 . The RT-PCR reactions program was completed with the following conditions: 95 °C for 30 s; 40 cycles of 95 °C for 5 s and 60 °C for 34 s; 95 °C for 15 s (*Cai et al., 2017*; *Cai et al., 2019*). The analyses were confirmed in triplicate. The relative expression level of each *AcoERF* gene was calculated based on the comparison threshold period ($2^{-\Delta\Delta Ct}$) method (*Pan et al., 2012*). The *protein phosphatase 2A* gene from pineapple was used as the reference gene (*Sang et al., 2013*). The final data were subjected to an analysis of variance test.

## RESULTS

### Genome-wide identification and characterization of pineapple ERF family

A total of 103 pineapple genes were identified as possible encoding proteins containing AP2/ERF domain (Fig. 1A). Among these, 27 genes containing two conserved AP2/ERF domains and 2 genes possessed a single AP2/ERF domain together with a B3 domain were assigned to AP2 and RAV family, respectively; Remaining 74 genes with a single AP2/ERF domain were grouped into ERF family, including 20 DREB and 54 ERF subfamily members. Earlier, the DREB subfamily members of pineapple had been divided into 5 groups: I, II, III, IV and V (*Chai et al., 2020*). To maintain uniformity, all the pineapple *ERF* genes were provisionally named as *AcoERF1- AcoERF74* according to the gene ID in ascending order
(Table S2). The full-length of the 74 *AcoERF* proteins range from 105 (*AcoERF36*) to 605 (*AcoERF17*) amino acid residues, while the coding sequence (CDS) length ranges from 318 to 1819 bp, and relative molecular weights (MW) range from 11.21 to 66.74 kDa. The predicted isoelectric points (*pI*) varied from 4.71 (*AcoERF18* and *AcoERF52*) to 9.63 (*AcoERF30*) (Table S2). Moreover, the Go annotation of *AcoERFs* included biological process (GO: 0006355) and molecular function (GO: 0003700) (Table S2). Within biological process, 72 *AcoERF* genes were associated with regulation of transcription and DNA-templated. For the molecular function categories, 73 *AcoERF* genes were involved in transcription factor activity and sequence-specific DNA binding.

## Multiple sequence alignment and phylogenetic analysis of pineapple *ERF* genes

Multiple sequence alignment of the 74 AcoERFs indicated that most ERF family members possessed conserved YRG and RAYD elements within the AP2/ERF domain region (Fig. S1). ERF subfamily proteins containing Ala at position 14 and Asp at position19. However, the residues at position 14 and 19 of DREB subfamily proteins is Val and Glu, respectively. Evolution of ERF TFs was further explored based on the phylogenetic tree constructed with pineapple and *Arabidopsis ERF* genes (Fig. 1B). Earlier, seven *Arabidopsis* ERF proteins having low homology with other member of ERF family, were assigned to group VI-L and Xb-L according to the analysis of gene structure and conserved motif (*Nakano et al., 2006*). In this study, these seven *Arabidopsis* ERF proteins were also used for phylogenetic reconstruction and formed group G and L. The *Arabidopsis* ERF proteins of group I and III were from the same group in previous study (*Nakano et al., 2006*). Therefore, group I and III in this study were renamed as C1 and C2, respectively. The result showed that the AcoERFs were divided into 13 groups, except for AcoERF21. Group I contained the most AcoERFs, accounting for 16.22% of total ERF genes. Followed by group J, which has 10 AcoERFs. However, group L has only one AcoERF.

## Gene structure and motif composition of pineapple ERF family

To gain more insight into the evolution and structural diversity of the ERF family in pineapple, we analyzed the gene structure and conserved motif of the *AcoERF* genes. As shown in Fig. 2, 49 *AcoERF* genes have no intron, accounting for 66.22% of the total number of *AcoERF* genes, and all genes of group J have no intron. Moreover, the number of introns in these genes varied from 1 to 16, and most *AcoERF* genes have 1 to 3 introns, whereas *AcoERF17* has the maximum number of introns. All genes of group H have 2 introns, except for *AcoERF8*. To further explore the divergence and functional relationship of AcoERF proteins, a total of 10 conserved motifs in the pineapple ERFs were identified by the MEME software, and the height of each letter in the logo was proportional to the conservation level of amino acid in all sequences analyzed (Fig. S2). As displayed in Fig. 3, motif 1 is present in every pineapple ERF protein, and almost all the proteins contain motif 2, except for AcoERF38 and AcoERF69. In addition, the motifs in different groups indicated that varying degrees of divergence among them. For example, motif 8 is unique to group H. Motif 6 is only present in group H and I. Motif 10 is only present in the

Peer**J**

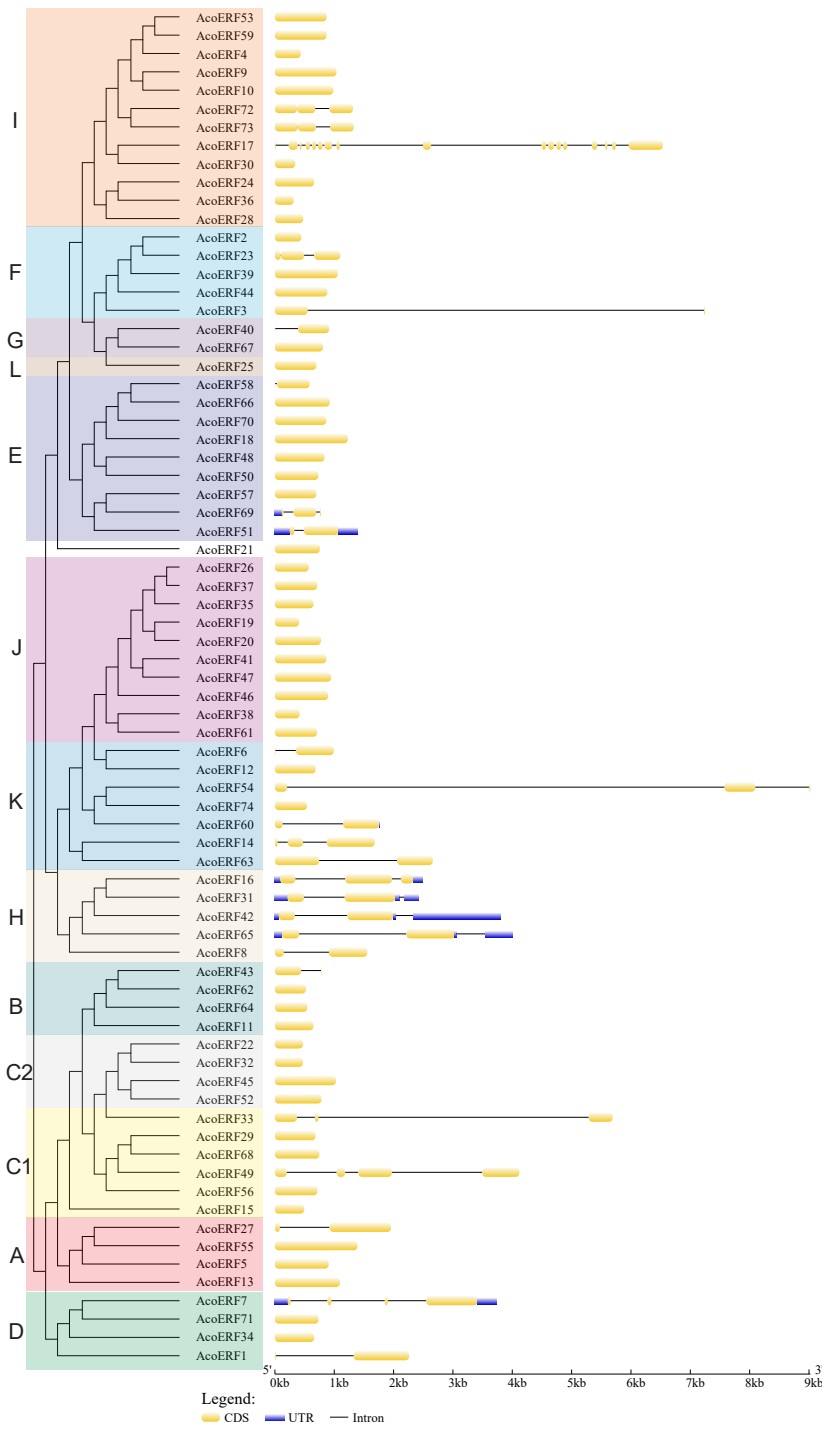

**Figure 2** **The exon-intron structure of *AcoERF* genes based on the evolutionary relationship.** The yellow round-corner rectangle represents exons, the black shrinked line represents introns, and the blue round-corner rectangle represents UTR.

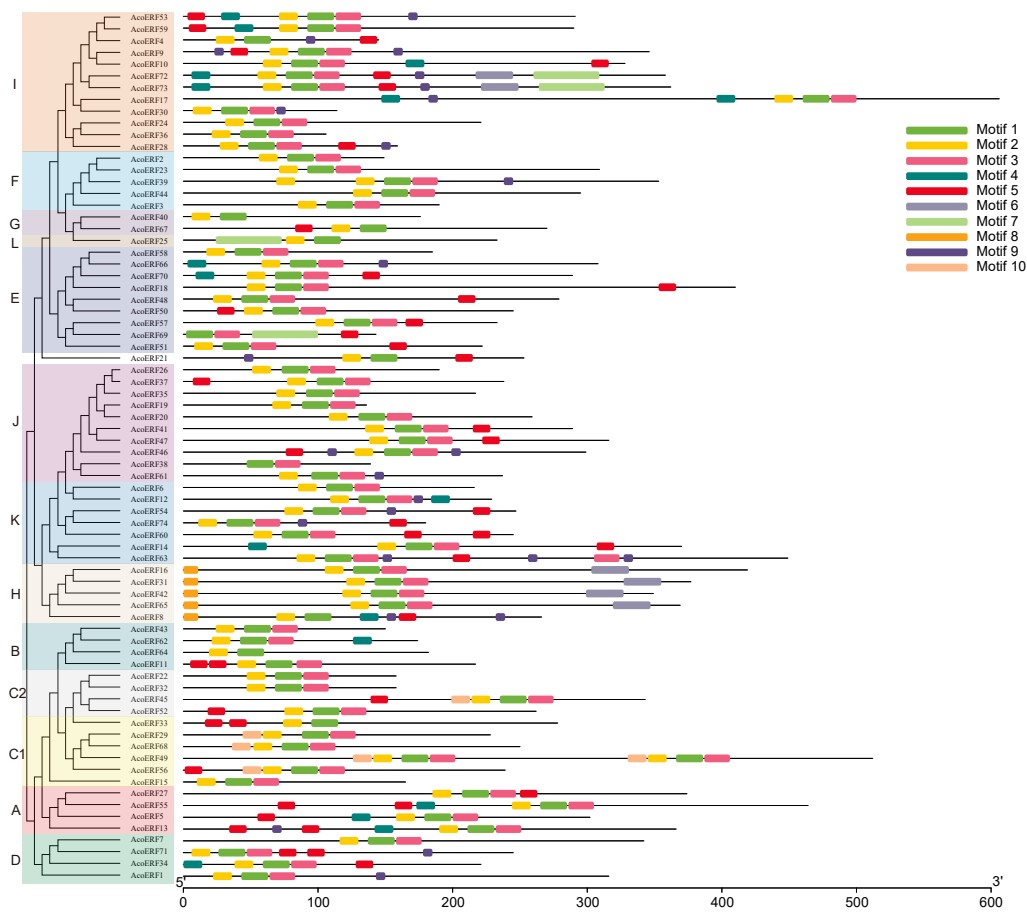

**Figure 3** **Motif distribution of AcoERF proteins.** The conserved motifs in the AcoERF proteins were identified with MEME software. Grey lines denote the non-conserved sequences, and each motif is indicated by a colored box numbered at the bottom. The length of motifs in each protein was presented proportionally.

members of group C1 and C2. In general, the pineapple ERF proteins in the same group usually contained similar motifs, which indicates that they may play similar roles in the development and growth of pineapple.

## Chromosomal distribution and synteny analysis of *AcoERF* genes

According to Fig. S3, 71 *AcoERF* genes were unevenly distributed on the 23 pineapple linkage groups (LG). Among them, LG02 contains the most diverse *AcoERF* genes, accounting for 12.16% of total genes, whereas LG04, LG14, LG19, and LG25 possessed only one *AcoERF* gene, respectively. Additionally, *AcoERF71*, *AcoERF73* and *AcoERF74* exist on unanchored scaffolds, so they could not be conclusively charted on any pineapple linkage groups (Table S2).

Segmental duplication provide an important mechanism for the expansion of gene families (*Lynch & Conery, 2000*; *Vision, Brown & Tanksley, 2000*; *Cannon et al., 2004*; *Zhang et al., 2018*). A total of 16 segmental duplication evens with 24 *AcoERF* genes

were found from pineapple genome, such as *AcoERF13/AcoERF5*, *AcoERF16/AcoERF31*, *AcoERF19/AcoERF35,* all of these were segmental duplicates (Fig. 4, Table S3), and these segmental duplication pairs play crucial roles in the expansion of ERF family in pineapple genome. In order to further study the phylogenetic and evolutionary trait of ERF family in pineapple, we set up 2 comparative syntenic maps of pineapple associated with 2 representative plant, including *Arabidopsis* and rice (Figs. 5A, 5B). According to the syntenic results, there are 37 and 44 orthologous gene pairs in *Arabidopsis* and rice, respectively (Tables S4 and S5). Some *AcoERF* genes had multiple orthologous gene pairs (one pineapple gene associated with multiple *Arabidopsis* or rice genes), such as *AcoERF62* associated with *AtERF008/019/020*, and *AcoERF29* associated with *OsERF025/026/027*, suggesting these genes might play crucial roles in the evolution of ERF family. Some *AcoERF* genes could be found orthologous gene pairs in *Arabidopsis* and rice, such as *AcoERF15*, *AcoERF22* and *AcoERF32*, suggesting that these genes might already exist before speciation. Some *AcoERF* genes only existed orthologous gene pairs between pineapple and rice, such as *AcoERF15*, *AcoERF22* and *AcoERF32*, suggesting that the appearance of these orthologous gene pairs before the divergence of monocotyledonous and dicotyledonous plants. In order to further understand the evolution of *ERF* gene family, the *Ka/Ks* ratios of the *ERF* gene pairs were calculated. The *Ka/Ks* ratio can represent different selection categories for duplication genes, such as the *Ka/Ks* >1 indicates positive selection , the *Ka/Ks* = 1 indicates neutral evolution, and the *Ka/Ks* <1 indicates negative selection (Li et al., 2019). According to the results, the *Ka/Ks* ratio of most orthologous *ERF* gene pairs are less than 1, indicating that they might have experienced strong negative selection pressure during the evolution of pineapple.

## Expression patterns of *AcoERF* genes in different tissues of pineapple

The expression profiles of 74 *AcoERF* genes in various pineapple tissues at different development stages was performed using RNA-seq expression data from MD2 pineapple plants recently published (*Ming et al., 2015*). According to the result, a total of 70 *AcoERF* genes were expressed in different tissues, whereas 4 *AcoERF* genes (*AcoERF22*, *AcoERF32*, *AcoERF43*, *AcoERF72*) were not found in the RNA-seq libraries (Fig. 6, Table S6). 22 genes were expressed in all tested tissues, in which 5 genes (*AcoERF13*, *AcoERF16*, *AcoERF31*, *AcoERF42*, *AcoERF65*) showed relative high expression levels (value > 10), suggesting these genes might played the indispensable roles in pineapple development. Moreover, 8, 16, 6, 9 genes showed high expression levels at all selected stages in ovule, sepal, stamen and petal, respectively; 16, 13, 12 genes showed high expression levels in leaf, flower and root, respectively; 10 genes showed high expression levels in fruit at all detected stages, indicating that these genes might be involved in the growth of relative tissues. Additionally, *AcoERF42* and *AcoERF65* showed the highest expression levels in fruit, suggesting that these genes might play positive roles in the growth and development of fruit.

## Identification of *cis*-elements in *AcoERF* promoters

In order to determine the response mechanisms of *AcoERFs*, 13 stress-related *cis*-elements were found in pineapple ERF promoters, such as low temperature response elements

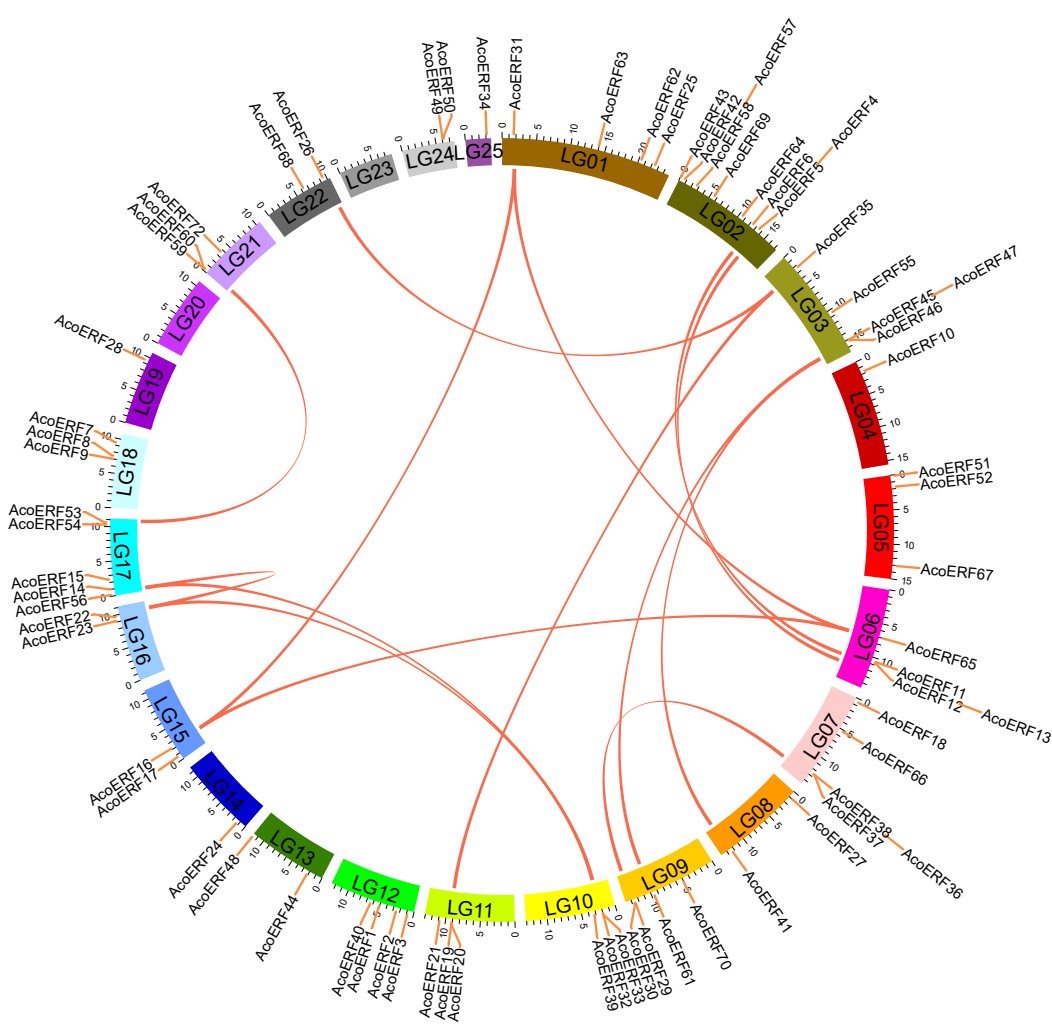

**Figure 4** **Schematic representations for the chromosomal distribution and interchromosomal relationships of pineapple *ERF* genes.** The red lines indicate duplicated ERF gene pairs, and the chromosome number is indicated at the bottom of each chromosome.

(LTR), dehydration responsive elements (DRE) and defense and stress responsive elements (TC-rich) (Table S7). LTRs are involved in low-temperature stress response (*Choudhury et al., 2008*); DRE and TC-rich repeats play roles in dehydration, low temperature, and salt stress response (*Diazdeleon, Klotz & Lagrimini, 1993*; *Germain et al., 2012*). According to the results, all *AcoERF* genes contain more than two *cis-elements* in their promoters. The promoter of *AcoERF23* only contained two *cis*-elements, whereas the most ten in the promoter of *AcoERF46*. In addition, at least one ABRE was present in 85.13% (63 out of 74) *AcoERF* gene promoters. The analysis of the 13 *cis*-elements suggested that most of *AcoERF* genes could response to different stress conditions.

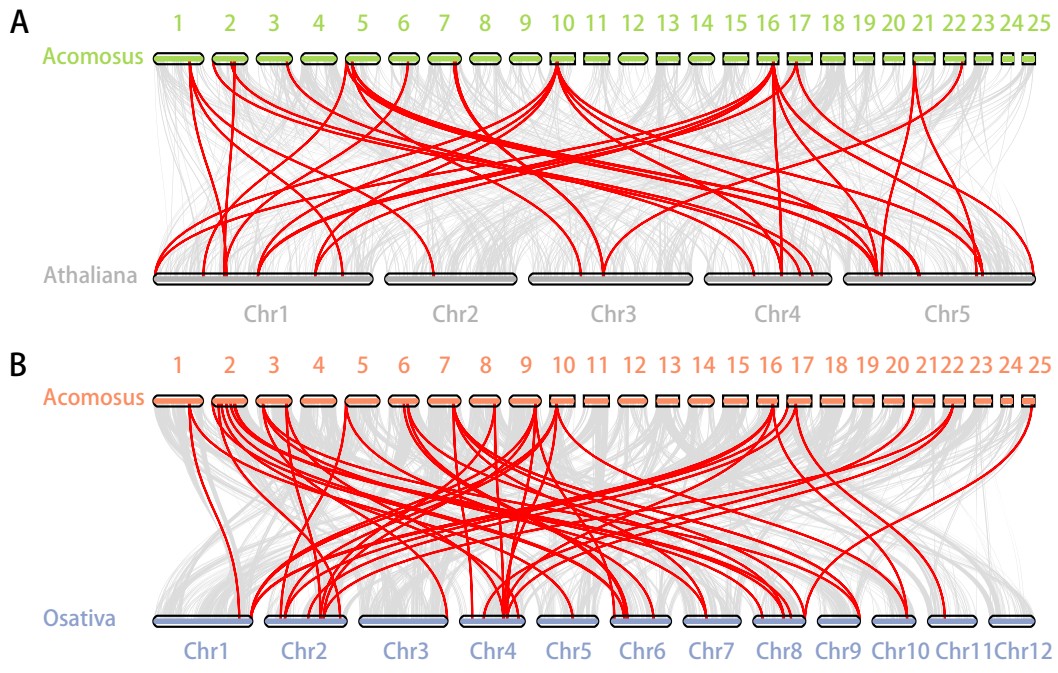

**Figure 5** **Synteny analysis of *ERF* genes between pineapple and two representative plant species.** (A) Synteny analysis of *ERF* genes between pineapple and *Arabidopsis*; (B) Synteny analysis of *ERF* genes between pineapple and rice. Gray lines in the background indicate the collinear blocks within pineapple and other plant genomes, while the red lines highlight the syntenic *ERF* gene pairs.

## Expression profiles of *AcoERF* genes under various abiotic stresses

According to previous studies, some *ERF* genes were involved in abiotic stress response in various plants, such as *Arabidopsis* (*Sakuma et al., 2002*), grape (*Zhuang et al., 2009*), tomato (*Sharma et al., 2010*), peach (*Amygdalus persica* L.) (*Zhang et al., 2012*). However, there was no report about the function of ERF genes in response to abiotic stress in pineapple. To investigate the roles of AcoERF genes in abiotic stress response, 9 *AcoERF* genes in roots and leaves were selected for functional verification. RT-PCR experiments were performed to analyze the expression patterns in response to different abiotic stress treatments, including cold, drought and salt stress (Figs. 7–9, Tables S8–S10). Overall, we found that the expression of these genes was influenced by the treatments. Among these treatments, some *AcoERF* genes in different pineapple tissues were significantly induced by the treatments, such as *AcoERF36 AcoERF45*, and *AcoERF67*. Interestingly, the expression patterns of AcoERF genes from root and leaf were different under the same treatments, such as *AcoERF13*, *AcoERF16* and *AcoERF42* under cold stress, *AcoERF7*, *AcoERF36* and *AcoERF60* in response to drought and salt stress, indicated that these genes may have different functions in various tissues of pineapple. Under cold and drought stress, the expression level of most *AcoERF* genes in leaves was higher than that in roots, while it was the opposite under salt stress, suggesting that *AcoERF* genes in root were sensitive to salt stress. Besides, most of *AcoERF* genes had maximal expression before 12 h in drought and
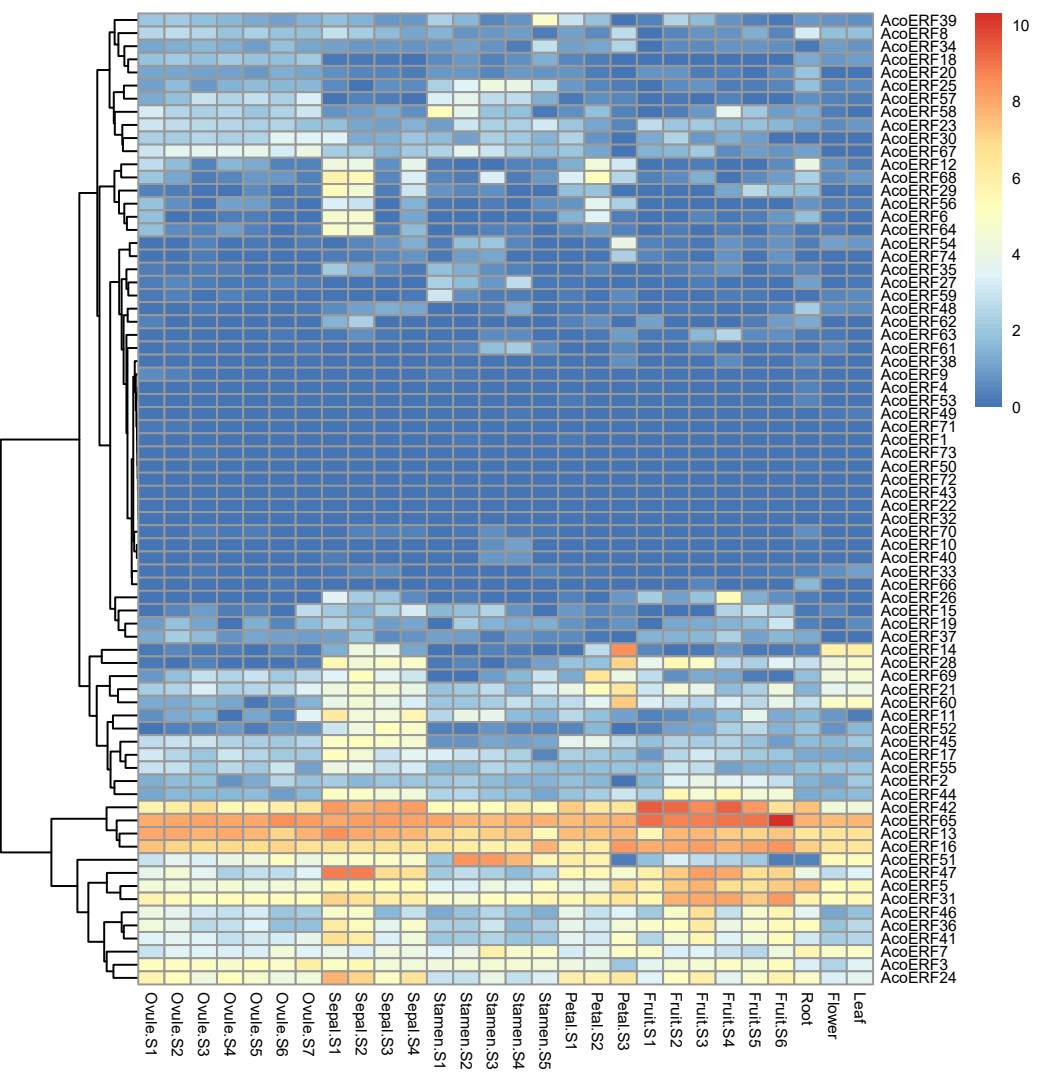

**Figure 6 Expression profiles of 74 *AcoERF* genes in different tissues and stages of pineapple.** Different colors in map represent gene transcript abundance values as shown in bar by the side of the figure.

salt stress, indicating these genes might play key roles in the early stage of drought and salt stress responses.

## DISCUSSION

The ERF family is one of the most important transcription factor families, and it belongs to AP2/ERF super-family, which also contains AP2 and RAV family (*Sakuma et al., 2002*; *Nakano et al., 2006*). It has been reported that the member of ERF family plays a crucial role in the growth and development of various plants (*Fits & Memelink, 2000*; *Banno et al., 2001*; *Yu et al., 2012*; *Muller & Munnebosch, 2015*). Previous studies have identified 120 ERF family members in soybean (*Li et al., 2005*), 103 in cucumber (*Hu & Liu, 2011*), 85 in tomato (*Sharma et al., 2010*), 139 and 122 in rice and *Arabidopsis*, respectively

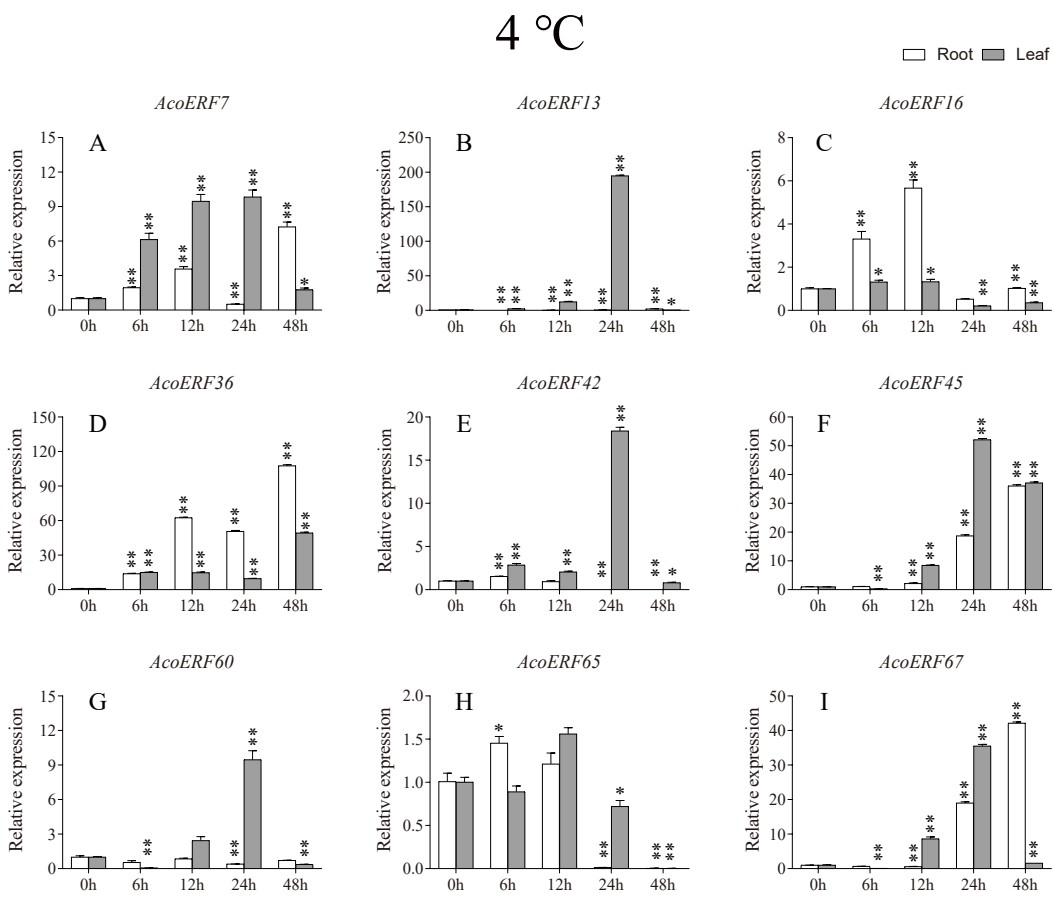

**Figure 7** **Expression profile of nine selected *AcoERF* genes in response to cold stress treatment.** The expression level of the 9 *AcoERF* genes (A–I) were gained by qRT-PCR. Error bars indicate the standard deviation. Asterisks on top of the bars indicating statistically significant differences between the stress and counterpart controls (\*P < 0.05, \*\*P < 0.01).

(*Nakano et al., 2006; Sharoni et al., 2011*), but the identification of pineapple ERF gene family has not been reported. Pineapple has great economic and research value, so it is necessary to investigate the function of ERF family in pineapple. In this study, 103 candidate *AP2/ERF* genes were identified in pineapple genome, including 27 AP2 family members, 2 RAV family members and 74 ERF family members (Fig. 1A). Compared with soybean, cucumber, tomato, rice and *Arabidopsis*, the pineapple ERF family is relatively small, indicating that some ERF members of pineapple may be lost during the evolution of species. To reveal the phylogenetic relationship of pineapple ERF family, a phylogenetic tree was constructed, 74 pineapple ERF family members were divided into 13 groups (Fig. 1B).

Gene structure analysis plays a crucial role in revealing the function of genes. Here, our results suggested that *AcoERFs* possess introns with number varying from 0 to 16, and the ERF members within the same group exhibited similar gene structure (Fig. 2). 66.22% *AcoERF* genes had no introns, which is similar to the status in *Arabidopsis*, cucumber and tartary buckwheat (*Nakano et al., 2006; Hu & Liu, 2011; Liu et al., 2019*). Presence of long,

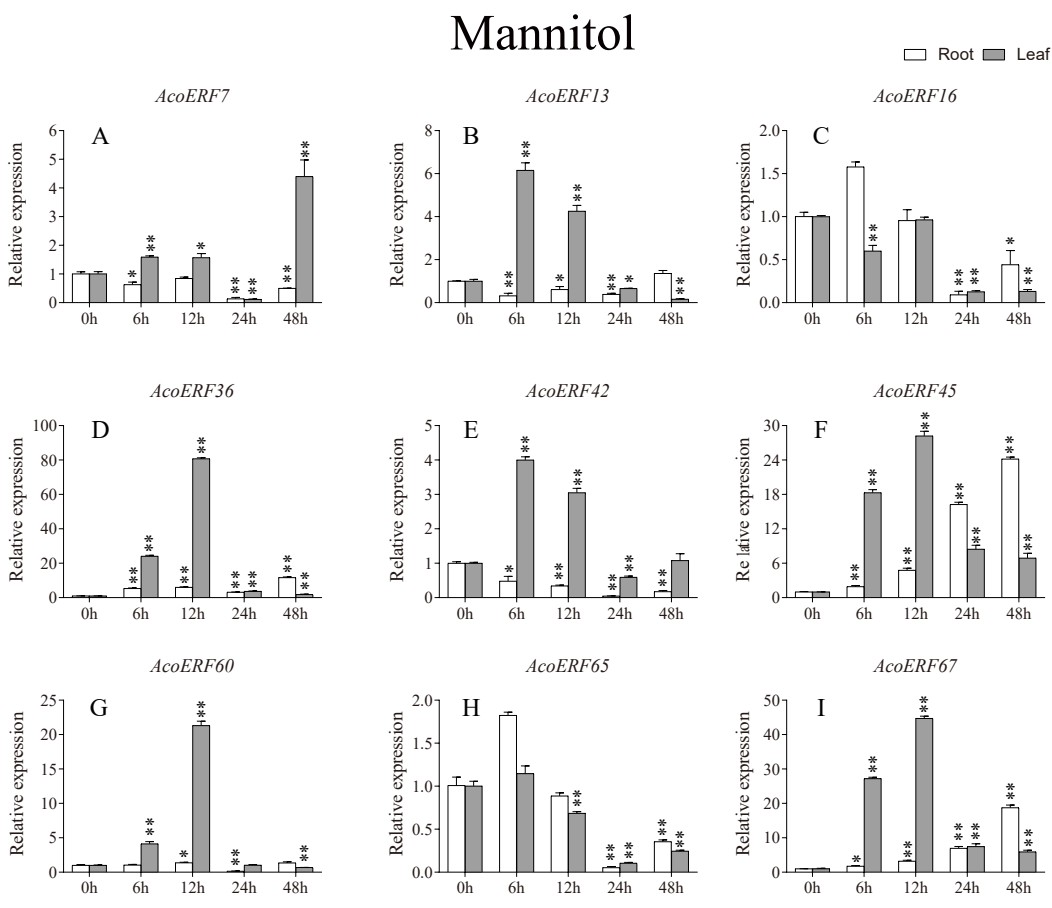

**Figure 8** **Expression profile of nine selected *AcoERF* genes in response to drought stress treatment.** The expression level of the 9 *AcoERF* genes (A–I) were gained by qRT-PCR. Error bars indicate the standard deviation. Asterisks on top of the bars indicating statistically significant differences between the stress and counterpart controls (*$P < 0.05$, **$P < 0.01$).

multiple intros can delay transcriptional output, which may suppress the expression of genes under adverse conditions. Conversely, the genes with small or fewer introns may have efficient expression in response to stress environments (*Jeffares, Penkett & Bahler, 2008*; *Heyn et al., 2015*). Hence, a large number of intron-less *AcoERF* genes may react rapidly when the external environment changes. For instance, *AcoERF13*, *AcoERF36*, *AcoERF45* and *AcoERF67* had no introns, and they had efficient expression under various stresses within 48 h (Figs. 7–9, Tables S8–S10).

The domains and motifs of transcription factors play essential roles in proteins interaction, transcriptional activity and DNA binding (*Liu, White & Macrae, 1999*). Here, a total of 10 conserved motifs in the pineapple ERFs were identified (Fig. 3). Different types and numbers of motifs in the pineapple ERF proteins could affect the diversity of gene function. Motif 1, 2 and 3 correspond to the AP2/ERF domain region, and were highly conserved in pineapple ERF family members. Motif 4 is present in several AocERFs and specifically rich in glutamine (Q). The presence of poly (Q) motif is related

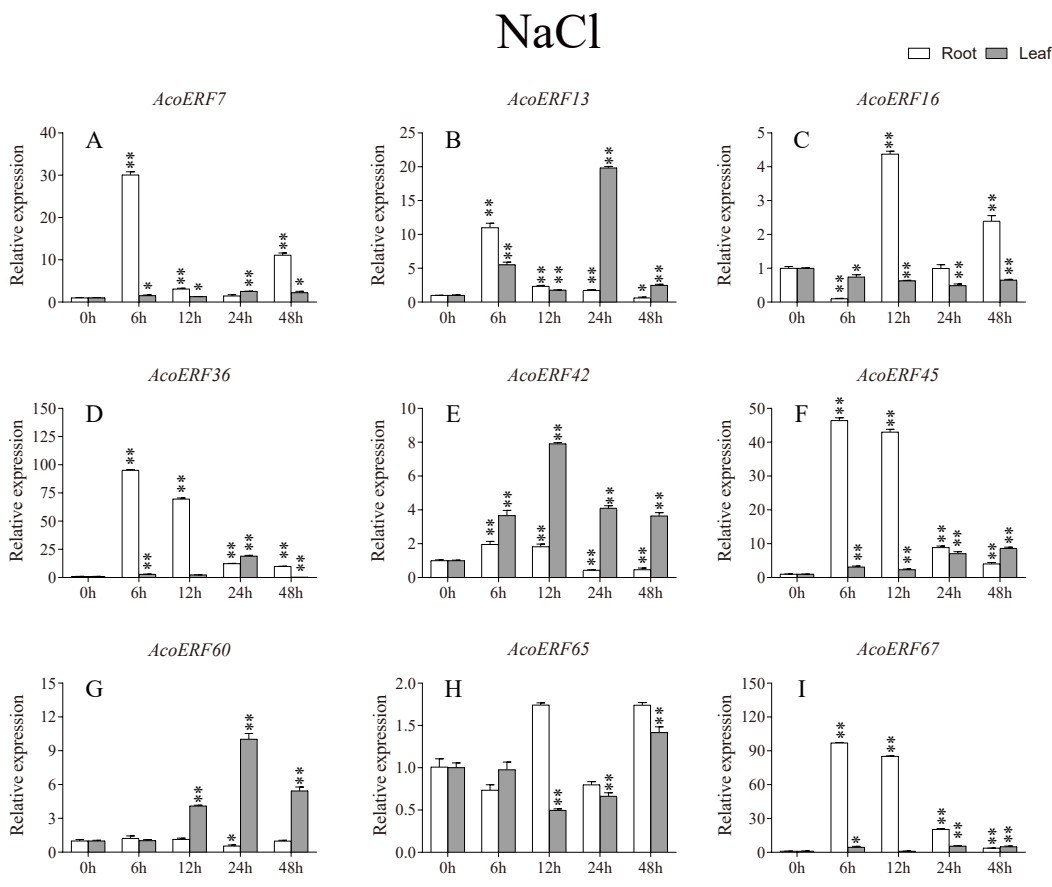

**Figure 9** **Expression profile of nine selected *AcoERF* genes in response to salt stress treatment.** The expression level of the 9 *AcoERF* genes (A–I) were gained by qRT-PCR. Error bars indicate the standard deviation. Asterisks on top of the bars indicating statistically significant differences between the stress and counterpart controls (*$P < 0.05$, **$P < 0.01$).

to protein aggregation and can stabilize protein-protein interactions (*Liu et al., 2011*; *Schaefer, Wanker & Andradenavarro, 2012*). Motif 5 containing multiple serine residues, and the poly serine repeats role as flexible linker, as well as an important site of several posttranslational modifications (*Uversky, 2015*). Therefore, AcoERFs containing motif 5 may paly crucial roles in various pathways. Motif 8 is specifically detected in group H, and the "MCGGAI" residues were highly conserved in the N-terminal region of the motif 8. It has been reported that the overexpression of the ERF genes with the "MCGGAI" motif can improve plant tolerance under hypoxia stress (*Xu et al., 2006*). Besides, the genes containing "MCGGAI" motif is also involved in ethylene transcription activation (*Buttner & Singh, 1997*). Although the function of some conserved motifs is still unknown, they may also be involved in transcriptional regulation. In general, the ERF family members within the same group shared similar gene structures and motif compositions, which suggests that they may have similar roles in plant development and growth.

Segmental duplication occurs frequently in plants since most plants are diploidized polyploids (*Zhu et al., 2014*). In this study, a total of 16 segmental duplication evens with

24 *AcoERF* genes were found from pineapple genome (Fig. 4, Table S3). The diversity of gene combinations may be one of the factors that make the regulatory relationship complicated (*Zhang et al., 2018*; *Zhang et al., 2020*). Polyploidization, a widespread phenomenon among plant, is considered an important process of plant speciation and evolution, and the formation of polyploid includes hybridization and genomic doubling process before or after hybridization (*Kohler, Scheid & Erilova, 2010*; *Zeng et al., 2020*). Various *Arabidopsis* and rice *ERF* genes have been reported to be involved in the regulation of plant stress tolerance. For instance, *At5g52020* and *OsDREB1F* have been shown to play crucial roles in high-salt, low-temperature and drought stresses (*Nakano et al., 2006*). The synteny analysis could reveal the functional and evolutional connections between two species. Here, 15 pineapple *ERF* genes and 25 *Arabidopsis ERF* genes, 23 pineapple and 32 rice *ERF* genes were identified as orthologous gene pairs, and different kinds of syntenic orthologous gene pairs could be investigated the evolutionary process of ERF family members in pineapple (Figs. 5A, 5B, Tables S4– S5). Some *ERF* genes did not find any orthologous gene pairs, which can be attributed to the rearrangement or fusion of chromosomes during their evolution (*He et al., 2014*; *Zhang et al., 2018*).

The analysis of gene expression patterns can be preliminarily predicted the function of genes (*Peng et al., 2015*; *Su et al., 2017*; *Zhang et al., 2018*). In this study, 5 genes (*AcoERF13*, *AcoERF16*, *AcoERF31*, *AcoERF42*, *AcoERF65*) showed high expression levels in all selected pineapple tissues (Fig. 6, Table S6), indicating that *ERF* genes may play a crucial role in pineapple development. Some *AcoERF* genes showed high expression levels in fruit, such as *AcoERF16*, *AcoERF45*, *AcoERF62*, suggesting that these genes play important roles in fruit ripening. Moreover, some *AcoERF* genes are expressed in various tissues or diverse stages, indicating that these genes could be more stable than those that only expressed in specific tissues or one stage of an organ (*He et al., 2019*).

When plants encounter stress conditions, a series of cell activities and molecules reaction mechanisms can improve the resistance of plants (*Chinnusamy, Schumaker & Zhu, 2004*; *Mittler, 2006*). According to previous studies, some *ERF* genes were involved in various stresses responses in plants, such as high-salt, low-temperature and drought stress (*Ohmetakagi & Shinshi, 1995*; *Sakuma et al., 2002*; *Nakano et al., 2006*). The expression profile of *AcoERF* genes suggested that several genes may be involved in the mechanism of abiotic stresses response (Figs. 7–9, Tables S8–S10). Besides, *AcoERF16*, *AcoERF42* and *AcoERF65* came from the same group H, and *AcoERF16* and *AcoERF65* was segmental duplication gene pair. Although these three genes had similar gene structure and motif compositions, the expression patterns of *AcoERF16* and *AcoERF65* were more similar under different treatments. The differential expression indicated that segmental duplication might influence the expression of gene. Overall, the above findings provide foundation to further investigate the potential function of pineapple *ERF* genes. These analyses are not only helpful in selecting valuable candidate *ERF* genes for further functional studies but also has important implication for genetic improvement for agricultural production and stress tolerance in pineapple crop.

## CONLUSIONS

Based on genomic data of pineapple, a genome-wide identification of pineapple ERF family was performed, and 74 *AcoERFs* were identified. A comprehensive analysis of their phylogenetic relationships, gene structures and conserved motifs compositions showed high levels of similarity within the same group. Synteny analysis of ERF genes revealed the evolutionary characteristic of pineapple ERF family. The expression profile of *AcoERFs* verified their roles in responding to various abiotic stresses (cold, drought and salt stress). These results will help to further study the *ERF* genes and their role in abiotic and biotic stress tolerance for improve the agricultural productivity of pineapple crop.

## ACKNOWLEDGEMENTS

We would like to thank the reviewers for their helpful comments on the original manuscript.

### Funding

This work was supported by the National Natural Science Foundation of China (U1605212, 31761130074 and 31970333 to Yuan Qin; 31700279 to Hanyang Cai) and a Guangxi Distinguished Experts Fellowship to Yuan Qin. The funders had no role in study design, data collection and analysis, decision to publish, or preparation of the manuscript.

### Grant Disclosures

The following grant information was disclosed by the authors:
National Natural Science Foundation of China: U1605212, 31761130074 and 31970333.
Guangxi Distinguished Experts Fellowship.

### Competing Interests

The authors declare there are no competing interests.

### Author Contributions

- Youmei Huang, Yanhui Liu and Man Zhang performed the experiments, authored or reviewed drafts of the paper, and approved the final draft.
- Mengnan Chai performed the experiments, analyzed the data, prepared figures and/or tables, and approved the final draft.
- Qing He, Bello Hassan Jakada, Fangqian Chen, Huihuang Chen and Xingyue Jin analyzed the data, prepared figures and/or tables, and approved the final draft.
- Hanyang Cai and Yuan Qin conceived and designed the experiments, authored or reviewed drafts of the paper, and approved the final draft.

### Data Availability

The raw measurements are available as Supplementary Files.
## Supplemental Information

Supplemental information for this article can be found online at http://dx.doi.org/10.7717/peerj.10014#supplemental-information.

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
