# Peer review of "Genome-wide identification and expression analysis of the ERF transcription factor family in pineapple (Ananas comosus (L.) Merr.)"

_PeerJ, doi:10.7717/peerj.10014_

## Round 0.1 · original submission · Major Revisions

Bothe reviewers raised important concerns that must be addressed in a revised version.

·

Basic reporting

In this manuscript, Huang et al. described identification of the 74 ERF transcription factor in pineapple and their expression analysis. The manuscript is reasonably well written. Although the introduction provides sufficient information to understand the background, it fails to highlight importance of this study. The manuscript requires extensive formatting of the citations and references.
- Line 53: Plant growth, development and adaptation do depend on several molecular players and their dynamic interactions. Citing a recent and brief review discussing such aspects a would be helpful to the reader rather than citing a paper with narrow focus such as Rashid et al., 2012.
- Since the authors performed comparative analysis (with rice and Arabidopsis) to find role of segmental duplication in evolution of this gene family, a gentle introduction on lineage specific polyploidization events in monocots would be helpful.
- There are several places with wrong citation formatting. For example
line 120: citation for MUSCLE has been mentioned as (C, 2004). It should be cited as (Edgar RC., 2004). There are several such cases (line 128,140)which makes it difficult to follow the references.
- Similarly, the reference section needs careful formatting. In several places a "%" appears before the journal name. e.g %J Bioinformatics, %J Plant physiology.
- In general, the tables and figures are helpful and relevant.
- The phylogenetic trees should display support scores such as bootstrap values.
- The taxon labels in Figure-2 should be formatted like Figure-1. Use different colors for different species.

- The paragraph headings in the Result section is basically repeating the headings of respective paragraphs in Method section. This makes the manuscript unattractive to the reader. The paragraph headings should be modified to tell summary of the findings.

Experimental design

- Line 103: "AP2/ERF genome sequences of pineapple" is it genome or peptide sequences? If Pineapple AP2/ERF genes are already known, then why do authors trying to identify them?
- Line 110: Source of "Arabidopsis AP2/ERFs" should be mentioned.
- Line 136: Authors should describe a clear strategy used to identify segmental duplicates.
- Line 144: It is not cleat whether the authors obtained expression values from a previous RNA-seq analysis or downloaded the raw reads to estimate the expression values. If they obtained the previous estimated expression values they should mention if it is raw read counts or the normalized read counts.
- Line 150: "heatmap" should be "heatmaply R package".

Validity of the findings

- Nearly 27% of the genes identified in this study has been reported as DREB genes in a previous study from the same group(10.7717/peerj.9006). Often single-AP2 domain proteins are often grouped into ERF and DREB sub-families (Nakano etal.,Plant Physiology,2006; Licausi etal., New Phytologist, 2013). Provided there exists a report on DREB genes, authors should classify and discuss something in this direction.
- As mentioned in my previous point that a previous study has identified 20 ERFs as AcoDREB genes (AcoDREB1-20), this study again re-identifies them as AcoERF. Authors should address this issue and reconsider the naming.
- They can also check the phylogenetic tree and ensure that six DREB groups are clearly identifiable in Figure 1.
- Line 340: Do pineapple orthologs of AtERF28 (At5g52020) and OsERF027 (OsDREB1F) also show similar expression under stress conditions?

Reviewer 2 ·

Basic reporting

Please refer to overall general comments

Experimental design

Please refer to overall general comments

Validity of the findings

Please refer to overall general comments

Additional comments

The manuscript by Huang et al present the repertoire of AP2 family of DNA binding domain in the Pineapple genome. While the authors have attempted to examine their repertoire, the analysis falls incomplete in many ways.
Major issues:
Primarily the AP2 DNA binding domain is not defined properly. The AT-hook motif and the domain’s structure containing three strands and one helix. Where exactly the DNA binding interface is located. (See and follow PMID: 16040597). This Lacuna is reflected in the analysis as described below.
The phylogenetic classification into 12 subclades is not supported by confidence values. What exactly the sequence features of these 12 subclades? Do they vary in the DNA binding interface or where the variation occurs? How this map on to the structure? Does it fall on the strands or the terminal helical region?
For all that matter, first three figures can be combined into 1. Moreover, the conserved motifs as the authors describe in Figure 4, What are those motifs? They mention Figure S1. But how the reader will know which motif is what. At least the motifs have to be described at first place. Where they are located in the structure and what it can infer us? None of these are described in the text.
What exactly the conclusions are? Yes, there are these many genes and so on but how this advance the understanding and how it will be useful for further downstream analysis is perplexing. In all the authors should perform in-depth sequence-structure analysis to consider the manuscript for publication.

---

## Round 0.2 · Minor Revisions

Before I accept the work, please address the following comments from Gerard Lazo, a Section Editor who has looked at your submission:

> The 74 ERFs were broken down into 13 groups and these were extrapolated into expression in different tissues and developmental stages. As the study attempts to detail expression based on tissue and developmental stages it would be important to add gene ontological terms to the sequences; none of this is seen anywhere in the manuscript. Journal manuscripts are often scanned by text-mining software that locates and extracts core data elements, like gene function.
>
> Adding standard ontology terms, such as the Gene Ontology (GO, geneontology.org) or others from the OBO foundry (obofoundry.org) can enhance the recognition of your contribution and description. This will also make human curation of literature easier and more accurate. None of this was visible.
>
> Tables should be included which link the proper GO:123456 numerical annotations to the gene IDs for the biological, cellular, and molecular terms. These are important for this study to be used for comparative applications with other related systems. There was no attempt to provide this so I would consider this manuscript as requiring moderate modifications.

·

Basic reporting

Please refer to the general comment section.

Experimental design

Please refer to the general comment section.

Validity of the findings

Please refer to the general comment section.

Additional comments

The authors have answered all of my concerns and have included most of my suggestions in manuscript. The MS can be accepted for publication.

---

## Round 0.3 · accepted · Accept

The manuscript has been improved after the authors revised the document following reviewers' and editors' suggestions. I recommend its acceptance in the current form.